# Molecular Typing and Antimicrobial Susceptibility Profiles of *Streptococcus uberis* Isolated from Sheep Milk

**DOI:** 10.3390/pathogens10111489

**Published:** 2021-11-16

**Authors:** Nives Maria Rosa, Ilaria Duprè, Elisa Azara, Carla Maria Longheu, Sebastiana Tola

**Affiliations:** 1Istituto Zooprofilattico Sperimentale della Sardegna G. Pegreffi, 07100 Sassari, Italy; rosa.nives@izs-sardegna.it (N.M.R.); idupre@agrisricerca.it (I.D.); elisa.azara@izs-sardegna.it (E.A.); carla.longheu@izs-sardegna.it (C.M.L.); 2Agris Sardegna, Servizio per la Ricerca nelle Produzioni Animali, Loc. Bonassai, SS291 km 18.600, 07100 Sassari, Italy

**Keywords:** *Streptococcus uberis*, mastitis, typing, antimicrobial susceptibility, resistance genes

## Abstract

Intramammary infections are a major problem for dairy sheep farms, and *Streptococcus uberis* is one of the main etiological agents of ovine mastitis. Surveys on antimicrobial resistance are still limited in sheep and characterization of isolates is important for acquiring information on resistance and for optimizing therapy. In this study, a sampling of 124 *S.* *uberis* isolates collected in Sardinia (Italy) from sheep milk was analyzed by multilocus-sequence typing (MLST) and pulsed field gel electrophoresis (PFGE) for genetic relatedness. All isolates were also subjected to antimicrobial susceptibility analysis by the disk diffusion test using a panel of 14 antimicrobials. Resistance genes were detected by PCR assays. MLST analysis revealed that the isolates were grouped into 86 sequence types (STs), of which 73 were new genotypes, indicating a highly diverse population of *S. uberis*. The most frequently detected lineage was the clonal complex (CC)143, although representing only 13.7% of all characterized isolates. A high level of heterogeneity was also observed among the *Sma*I PFGE profiles, with 121 unique patterns. Almost all (96.8%) isolates were resistant to at least one antimicrobial, while all exhibited phenotypic susceptibility to oxacillin, amoxicillin-clavulanic acid and ceftiofur. Of the antimicrobials tested, the highest resistance rate was found against streptomycin (93.5%), kanamycin (79.8%) and gentamicin (64.5%), followed by novobiocin (25%) and tetracycline-TE (19.3%). Seventy-four (59.7%) isolates were simultaneously resistant to all aminoglycosides tested. Seventeen isolates (13.7%) exhibited multidrug resistance. All aminoglycosides-resistant isolates were PCR negative for *aad*-6 and *aph*A-3′ genes. Among the TE-resistant isolates, the *tet*M gene was predominant, indicating that the resistance mechanism is mainly mediated by the protection of ribosomes and not through the efflux pump. Three isolates were resistant to erythromycin, and two of them harbored the *erm*B gene. This is the first study reporting a detailed characterization of the *S. uberis* strains circulating in Sardinian sheep. Further investigations will be needed to understand the relationships between *S. uberis* genotypes, mastitis severity, and intra-mammary infection dynamics in the flock, as well as to monitor the evolution of antimicrobial resistance.

## 1. Introduction

Mastitis is the most prevalent and costly disease affecting dairy sheep. Sardinia, an island located in the middle of the Mediterranean Sea, has approximately 3.5 million milking Sarda sheep, corresponding to half of the total Italian stock. Shepherding is a relevant part of the regional economy, particularly concerning pecorino cheese production. Therefore, udder health is a critical factor, and control of intra-mammary infections (IMI) is consequently of the greatest importance for dairy farmers. Infectious mastitis outbreaks of small ruminants are usually caused by Gram-positive bacteria, mostly *Staphylococcus* and *Streptococcus* species [1,2,3,4]. In a recent study concerning the identification of *Streptococcus* isolates, we found that *Streptococcus* (*S.*) *uberis* is the pathogen most frequently isolated from sheep and goat milk samples [5]. *S. uberis* is considered an environmental pathogen since it is commonly isolated from the environment, milking machines, milkers’ hands, and the skin of the teats. Mastitis may therefore arise when the mammary gland is exposed to high loads of this bacterial species.

Genotyping of *S. uberis* isolated from milk samples is an essential tool in mastitis epidemiology and contributes to our understanding of the pathogen’s dissemination. At present, very few studies report the genotypic characterization of *S. uberis* isolated from the milk of sheep with infectious mastitis [6]. On the other hand, several molecular methods have been applied for typing *S. uberis* isolated from bovine milk, including multi locus sequence typing (MLST) and pulsed field gel electrophoresis (PFGE) [7,8,9]. PFGE is the most discriminatory method in outbreak analysis [10], whereas MLST is considered the method of choice to investigate populations and population dynamics on a local and global scale [11].

Antimicrobial therapy continues to play an important role in controlling both clinical and sub-clinical streptococcal mastitis. For preventing the emergence and selection of multidrug-resistant pathogens, it is essential to monitor the trends in antimicrobial resistance (AMR) among bacteria. The World Health Organization (WHO) has declared that AMR is one of the top 10 global public health threats facing humanity. The main AMR drivers include the misuse and overuse of antimicrobials; lack of access to clean water, sanitation, and hygiene for both humans and animals; poor infection and disease prevention measures, and control in health-care facilities and farms.

The study aimed to analyze the genetic diversity and the antimicrobial susceptibility profiles of *S. uberis* field isolates collected during a 7-year period from small ruminants in Sardinia, Italy.

## 2. Results

### 2.1. Streptococcus uberis Genotyping

All 124 *S. uberis* were typed by both MLST and PFGE analysis. MLST analysis identified 86 different allelic combinations; among these, 13 were STs present in the database while the remaining 73 (84.9%) STs were found for the first time in this study. The ST was considered new if the allelic combination did not overlap with known STs in the *S. uberis* MLST database. The list of the new IDs assigned to the *S. uberis* isolates, marked as IZSar, as well as country, year of isolation, MLST profile, ST, and clonal complex (CC) are displayed in https://pubmlst.org/bigsdb?db=pubmlst_suberis_isolates (accessed on 29 April 2021). The most frequently detected STs were ST808, ST1177, ST1192 and ST1247 with 4 isolates each, followed by ST562, ST1174, ST1176, ST1182, ST1188 and ST1242 with 3 isolates (Appendix A). The highest number of new alleles was detected for *tdk* with 10 new alleles, followed by *yqi*l and *arc*C with 7 and 6 alleles, respectively (Appendix A). CC143, with 8 different STs grouping 17 isolates, was the most abundant clonal complex among the ovine *S. uberis* isolates, followed by CC86 (four STs and 5 isolates) and CC5 (one ST and 1 isolate) (Appendix A). The remaining STs did not belong to any CC. The goeBURST full MST analysis identified a total of 20 possible ST founders and 4 within CC143 (ST294, ST808, ST1192 and ST1195). The founder ST294 contained 4 other STs: 2 with a single locus variant (SLV) and 2 with double locus variants (DLVs). The founder ST808 had 6 STs: 2 with a SLV, 3 with DLVs and 1 with triple locus variants (TLVs) (Figure 1). Figure 2 illustrates the relationship between the 86 STs of this study and those (*n* = 51) extracted from the *S. uberis* MLST database and related to other Italian regions. These *S. uberis* were mainly isolated from bovine mastitis. Unlike what has been found in ovine mastitis, the CC5 was predominant in *S. uberis* isolates from bovine mastitis (Figure 2). The goeBURST analysis showed that the majority of Italian cattle strains are grouped in the ST 1199 which is a SLV of the predicted founder ST808 (Figure 2).

All 124 *S. uberis* isolates were also typed by PFGE. Figure 3 shows the percentage similarity among patterns determined by cluster analysis in a dendrogram. A high level of polymorphism was observed among the *Sma*I profiles; in fact, a total of 121 pulsotypes (PTs) were unique within this dataset. By using a cut off similarity value of 75% in the UPGMA dendrogram, 68 *S. uberis* grouped into 29 PFGE clusters (from 1 to 29) whereas 56 isolates had a lower level of similarity in comparison to other isolates and were not assigned into clusters.

### 2.2. Antimicrobial Susceptibility and Resistance Genes

Almost all *S. uberis* isolates (120/124, 96.8%) were resistant to at least one antimicrobial. Resistance was exhibited to all the antimicrobials tested with the exception of AMC, OX and EFT. Of the antimicrobials tested, the highest resistance rate was recorded against S (116/124, 93.5%), K (99/124, 79.8%) and CN (80/124, 64.5%), followed by NV (31/124, 25%), and TE (24/124, 19.3%). Seventy-four (59.7%) isolates were simultaneously resistant to all aminoglycosides. Seventeen isolates (13.7%) were MDR: ten were resistant to three different classes, six to four different classes and one to five different classes (P, TE, S, SXT and NV) (Figure 4). No specific correlation was found between ST/CC/ PFGE and resistance to aminoglycosides or other antimicrobials. On the basis of inhibition zone diameters, established by guidelines of CLSI, many isolates presented intermediate values with respect to E and TE: 17 (13.7%) and 39 (31.5%), respectively. All of them were analyzed by PCR for the corresponding resistance genes. Of the three E-resistant isolates, two harboured *erm*B gene while one carried *erm*C gene. The *erm*A, *erm*TR and *mef*A genes were not detected among the analyzed isolates. All E-intermediate isolates were PCR negative. Among the twenty-four *S. uberis* isolates resistant to TE, ten were positive for *tet*M, six for *tet*O, two for *te*tK and one for *tet*O. One isolate carried simultaneously *tet*K and *tet*M genes, one for both *tet*O and *tet*S, and another one for both *tet*k and *tet*O genes. Seven isolates were non-typeable using the primers for these *tet* genes. Among the 39 TE-intermediate isolates, 3 were PCR positive for *tet*K and 2 for *tet*M. One isolate possessed both *tet*K and *tet*M genes. Of the six P-resistant isolates, five harbored the *bla*Z gene, encoding β-lactamase, which is responsible for enzymatic hydrolysis of the ring of β-lactam antibiotics. All aminoglycosides-resistant isolates were PCR negative for *aad*-6 and *aph*A-3′ genes.

## 3. Discussion

The present study contributes to understand the population structure of *S. uberis* involved in ovine mastitis in Sardinia, Italy. Several studies are available on *S. uberis* from dairy cows [8,9,12,13,14,15,16], while only limited reports are available on the genotyping and antimicrobial resistance of *S. uberis* from small ruminants [6,17]. In this work, which represents the first study dealing with molecular typing of ovine *S. uberis* in Italy, the genetic relatedness of isolates was established by using MLST and PFGE.

The data generated by the MLST analysis was used to implement the *Streptococcus uberis* database (https://pubmlst.org/organisms/streptococcus-uberis) (accessed on 29 April 2021) and to compare all isolates collected in Italy, even if of different animal origin. The MLST approach is well suited for such comparative studies; in addition to distinguishing continuous and new infections, it also allows to investigate the genetic lineage of the isolates. Our results showed that the 124 isolates were grouped into 86 STs, of which 73 were new STs. Among these, only ST294 (Appendix A) is common to the 14 ovine isolates from the Lazio region of Italy analyzed in a previous study [6]. ST294 is considered a possible ST founder by the goeBURST full MST algorithm (Figure 1). ST294, ST384, ST562, ST808, ST1112, ST1189, ST1192 and ST1195 were grouped into CC143, which represents the most widespread lineage in Sardinia, although including only 13.7% of all isolates considered in this study (Figure 2). CC86 and CC5 were detected at very low rates of 4% and 0.8%, respectively. Although there are no other data concerning *S. uberis* of small ruminants, lineages CC143 and CC86 were highly associated with bovine IMI in Portugal [18], India [19], Italy [6] and Australia [16], whereas CC5 is the most prevalent clonal complex among bovine mastitis isolates collected in the UK [20] and Swiss Midlands [7].

According to Tomita et al. [12], CC5 and CC143 are correlated with clinical and subclinical bovine mastitis and may represent lineages of virulent isolates, while isolates belonging to CC86 may be associated with low somatic cell count cows. This hypothesis is based on the detection of specific virulence factors in 95% of the isolates grouped in CC5 and CC143 and only in 25% in those belonging to CC86. Work is in progress for detecting virulence marker genes in our isolates.

In agreement with other authors [14,18,21] on molecular typing by PFGE, macrorestriction analysis revealed a high degree of genetic diversity even among *S. uberis* from small ruminant origin. Using a similarity value of ≥75%, the isolates were distributed in 29 small clusters. Two isolates, belonging to cluster 3, exhibited the same PFGE profile, and were collected from different animals of the same flock. Two other isolates, belonging to cluster 13, presented the same PFGE pattern. In this latter case, isolates were recovered from distinct sheep farms distant; about 23 km, as the crow flies. The last two isolates with the same PFGE pattern, belonging to cluster 19, were collected from two separate samplings, made 10 months apart, on the same sheep farm. The occurrence of isolates with identical PFGE profiles could be due to their transmission from sheep to sheep through the milking process (1st and 3rd cases) or to the commercial movement of infected animals between farms (2nd case). According to the criteria described by Tenover et al. [10], isolates showing indistinguishable macrorestriction patterns should be considered the same strain. However, in the comparison between PFGE and MLST, we found that, in the 1st case, the isolates presented different STs (ST1176 and ST1179), even if it concerned only the *yqi*L allele; in the 2nd case, STs also were different (ST1182 and ST1179) with 5 allelic variants of *ddl*, *gki*, *tdk*, *tpi* and *yqi*L genes. In the 3rd case, STs were identical. In light of these results, it would be correct to consider the same strain only the isolates belonging to the 3rd case. Whole genome sequencing (WGS) characterization will be required to shed further light on this hypothesis.

In the present study, we evaluated the AMR of the 124 *S. uberis* isolates to 14 antimicrobial agents. A recognized limitation of AMR tests is the lack of small ruminants-specific interpretive criteria to categorize isolates as susceptible, resistant, or intermediate. In this study, the cut-off values used are based on *Streptococcus/Staphylococcus* from human and bovine origin [22]. Using these breakpoints, a high percentage (59.7%) of isolates were found to be simultaneously resistant to the three aminoglycosides tested, with peaks of 93.5%, 79.8% and 64.5% for S, K and CN, respectively. High percentages of resistance to these antimicrobials using the same method were observed by other researchers [17,23]. An acquired high level of resistance to streptomycin can be explained by the extensive use of this antimicrobial in combination with penicillin in the treatment of ovine mastitis. However, we did not find the presence of genes associated with resistance to aminoglycosides (i.e., *aad*-6 and *aph*A-3’) among the isolates analyzed in our research. Taber et al. [24] hypothesized a “natural” resistance of streptococci to aminoglycosides resulting in a low level of intrinsic resistance to these drugs. Further studies are needed to determine the presence of other resistance determinants and the mechanism of gene transfer. In this work, 19.3% of *S. uberis* isolates were phenotypically resistant to tetracycline; this resistance mainly depends on ribosomal protection proteins, mediated by *tet*M gene (41.6%) or *tet*O (25%) determinants. This finding is consistent with some previous surveys on *Streptococcus* spp. collected from dairy cattle in Poland [25], in the United States [26] and China [27]. Therefore, it can be stated that the *tet*M gene is involved in tetracycline resistance not only in human streptococci [28] but also in those isolated from bovine and ovine mastitis. For macrolides, we investigated the presence of *erm*A, *erm*B, *erm*C, *erm*TR and *mef*A genes that codify methylase which reduce the binding of antibiotics to the target site. Only 3 *S. uberis* were E-resistant isolates: two harbored *erm*B while one carried *erm*C. These findings were consistent with previous studies, which found a predominance of *erm*B among bovine isolates [23,25,27]. A noteworthy result in our study was the high resistance (25%) of *S. uberis* isolates to novobiocin, which is not reported in the literature.

## 4. Materials and Methods

### 4.1. Bacterial Isolates and DNA Extraction

In this study, we analyzed 124 *S. uberis* isolated between January 2011–December 2017 from sheep milk collected in different provinces of Sardinia (Italy). The isolates belonged to a collection of *Streptococcus* species used for the preparation of inactivated autogenous vaccines, due to the presence of clinical or subclinical mastitis cases in the flock. Information about the type of mastitis was not provided by the farm veterinarian. With the exception of four isolates, all the others were from different dairy farms. Bacterial isolation and identification at the species level by PCR-RFLP and MALDI-TOF MS analysis, have been described in a previous study [5].

Bacterial DNA was extracted from all isolates and from *S. uberis* reference strain ATCC 700407 according to Onni et al. [29]. Briefly, isolates were grown in 5 mL Brain Heart Infusion broth (BHI, Oxoid Ltd., Basingstoke, UK) at 37 °C for 18 h with shaking. A 100 µL aliquot was pelleted by centrifugation at 7000 rpm for 2 min. The pellet was resuspended in 49.5 µL deionized sterile water supplemented with 0.5 µL lysostaphin (1 mg/mL) dissolved in 20 mM sodium acetate, and incubated at 37 °C for 10 min. After addition of 1 µL proteinase K (5 mg/mL) (Roche Diagnostics, Monza, Italy) and 150 µL of 0.1 M Tris-HCl (pH 7.5), samples were further incubated for 10 min. Finally, samples were boiled for 5 min and then stored at −20 °C.

### 4.2. MLST

Amplification of the seven housekeeping genes [carbamate kinase (*arc*C), D-ala-D-ala ligase (*ddl*), glucose kinase (*gki*), transketolase (*rec*P), thymidine kinase (*tdk*), triose phosphate isomerase (*tpi*) and acetyl CoA acetyl-transferase (*yqi*L)] was performed using the primer sequences as described by Coffey et al. [11]. Gene fragments were amplified by conventional PCR (GeneAmp PCR System 9700, Applied Biosystems, Foster City, CA, USA), then purified and sent to BMRGenomics (https://www.bmr-genomics.it/) (accessed on 28 September 2020) for DNA sequencing. The allelic profile of each isolate was identified and, where possible, strains were assigned to sequence type (ST) using the *S. uberis* MLST database (https://pubmlst.org/organisms/ streptococcus-uberis) (accessed on 28 September 2021). Alleles not identified as well as unknown allelic profile were submitted to the database and new STs were generated. Phyloviz version 2.0 (www.phyloviz.net) (accessed on 29 April 2021) and goeBURST algorithm [30] were used to visualize the data. The same algorithm was also used to compare our strains with other *S. uberis* of ruminant origin isolated in Italy, and present in the MLST database (accessed on 29 April 2021).

### 4.3. PFGE

All 124 *S. uberis* isolates were genotyped by PFGE. DNA was extracted with Bio-Rad CHEF genomic DNA plug kit (BioRad, Segrate, Italy) according to the manufacturer’s instructions. Each plug was digested with 20 U of *Sma*I (Roche) for 3 h at 25 °C and washed with 5 mL of TE buffer for 10 min at room temperature before being loaded on a 1% certified Megabase agarose (BioRad) gel. The lambda PFG ladder (New England Biolabs, Ipswich, MA, USA) was used as a molecular size standard. Electrophoresis was carried out in a contour-clamped homogeneous electric field (CHEF)- Mapper system (BioRad) at 14 °C in 0.5× TBE buffer. DNA fragments were separated after 18 h migration with 6 V/cm, 120° at pulse times of 10–45 sec. Gels were stained with ethidium bromide and photographed with Alliance LD2 gel documentation system (UVITEC, Cambridge, UK) in TIFF format. The *S. uberis* digitalized PFGE patterns were analyzed with the Gel Compar II software (Applied Maths, Sint-Martens-Latem, Belgium). Dendrograms were generated by the unweighted pair group method with arithmetic averages (UPMGA) using the Dice correlation coefficient with a position tolerance of 1.5%. Isolates with ≥90% similar profiles were considered to represent the same clone [10].

### 4.4. Antimicrobial Susceptibility Testing

Antimicrobial susceptibility testing was performed through the disc diffusion method on Mueller Hinton agar supplemented with 5% of defibrinated sheep blood using an inoculum corresponding to the 0.5 McFarland standard. The plates were incubated at 37 °C in an atmosphere of 5% CO_2_ for 24 h before measuring the zone of inhibition. The following antibiotic discs (Oxoid, Basingstoke, UK) were used: Ampicillin (AMP, 10 ug) penicillin (P, 10 IU), tetracycline (TE, 30 μg), streptomycin (S, 10 μg), kanamycin (K, 30 μg), gentamicin (CN, 10 μg), erythromycin (E, 15 μg), trimethoprim-sulphamethoxazole (SXT, 25 μg), amoxicillin-clavulanic acid (AMC, 30 μg), cephalothin (KF, 30 μg), oxacillin (OX, 5 μg), novobiocin (NV, 30 ug), ceftiofur (EFT, 30 ug) and pirlimycin (PIR, 2 ug). *S. pneumoniae* ATCC 49619 were used as the quality control strain. Isolates were classified as susceptible, intermediate or resistant based on inhibition zone diameters, according to guidelines of the Clinical and Laboratory Standards Institute [22]. For NV, EFT and PIR, the susceptibility breakpoints were based on *S. uberis* collected from bovine mastitis. For the remaining antimicrobials, the susceptibility categorization was based on human *Streptococcus*-derived breakpoints. Multidrug resistance (MDR) was defined as resistance to ≥3 antimicrobial classes.

### 4.5. Detection of Resistance Genes by PCR

Genes encoding resistance to aminoglycosides (*aad*-6 and *aph*A-3′), tetracyclines (*tet*O, *tet*K, *tet*M, *tet*L and *tet*S), macrolides (*erm*A, *erm*B, *erm*C, *erm*TR and *mef*A) and penicillins (*bla*Z) were investigated by PCR using the primers presented in Appendix A. PCRs were performed in a thermocycler (GeneAmp 9700, Applied Biosystems, Waltham, MA, USA) using the following program: initial denaturation for 5 min at 95 °C followed by 30 cycles of 1 min at 95°, 1 min at 37–58 °C (according to the annealing temperature for the individual primers; Appendix A) and 1 min at 72 °C with a final extension step of 10 min at 72 °C. PCR products were examined by electrophoresis in 1.5% agarose gels, stained with Sybr^®^Safe DNA gel stain (Invitrogen, Waltham, MA, USA) and visualized under a UV transilluminator. A DNA molecular weight Marker VIII (Roche) was used to determine the size of the amplification products. The following positive isolates/controls were included: *Staphylococcus haemolyticus* 772 (*erm*A), *Enterococcus faecalis* 8855 (*erm*B), *Staphylococcus haemolyticus* 15680 (*erm*C)*, Streptococcus pyogenes* ATCC 12344 *(ermT*R, *mef*A), *Staphylococcus aureus* 4438 (*tet*K), *Staphylococcus epidermidis* 1464 (*tet*L), *Staphylococcus aureus* 2412 (*tet*M), *Staphylococcus aureus* 4438 (*tet*O), *Lactococcus lactis* subsp. *lactis* ATCC 15577 (*tet*S), *Staphylococcus aureus* ATCC 33591 (*bla*Z), *Streptococcus oralis* ATCC 35037 (*aad*-6) and *Escherichia coli* ATCC 11775 (*aph*A-3′).

## 5. Conclusions

To sum up, genotyping by MLST and PFGE analysis indicates the epidemiological complexity of *S. uberis* isolated from ovine milk. The lack of a predominant strain and the demonstration that a wide variety of isolates are capable of infecting the mammary gland suggests that most of the *S. uberis* IMI cases are due to contamination of the mammary gland from the environment. Nevertheless, further research is needed to shed light on the possible transmission of some strains within or among flocks. The high resistance of the isolates toward aminoglycosides, novobiocin and tetracycline confirms that greater care should be taken when these antimicrobials are used in the veterinary practice. Continuous monitoring should be carried out to maintain an updated understanding of the level of antimicrobial resistance among dairy sheep in Sardinia.

## Figures and Tables

**Figure 1 pathogens-10-01489-f001:**
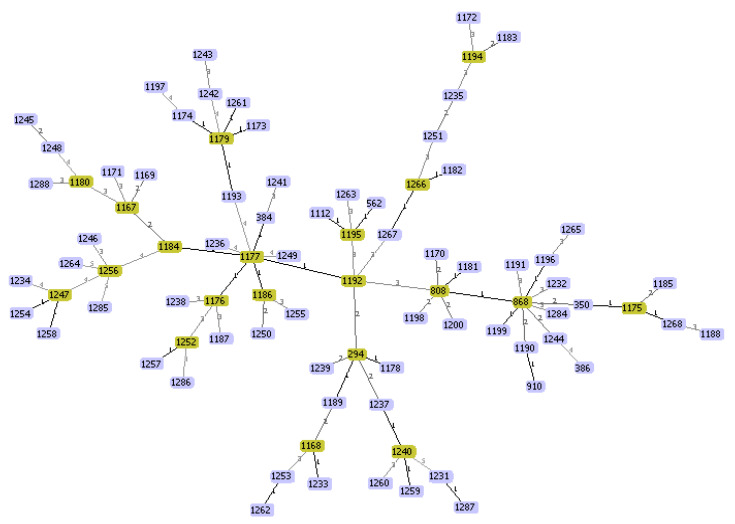
A complete minimum spanning tree (MST) based on allelic profiles of *Streptococcus uberis* isolated from ovine milk samples in Sardinia, Italy. The tree was generated by goeBURST full MST algorithm in PHYLOViZ 2.0 (http://www.phyloviz.net) (accessed on 29 April 2021). Numbers within the nodes indicate the ST. Yellow nodes correspond to the possible ST founders. Numbers on lines indicate locus variants between adjacent nodes.

**Figure 2 pathogens-10-01489-f002:**
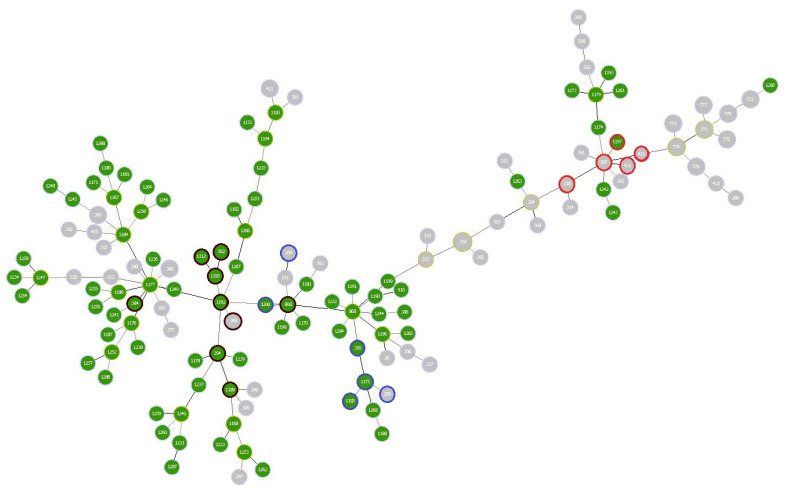
A complete minimum spanning tree (MST) of 137 *Streptococcus uberis* isolates from ovine (green) and bovine (grey) mastitis, collected in Italy. Isolates from bovine origin (*n* = 51) were downloaded from *S. uberis* MLST database (https://pubmlst.org/organisms/streptococcus-uberis) (accessed on 29 April 2021). Numbers within the nodes indicate the ST. The color of the circle border refers to the clonal complex (CC) found: dark brown (CC143), blue (CC86) and red (CC5).

**Figure 3 pathogens-10-01489-f003:**
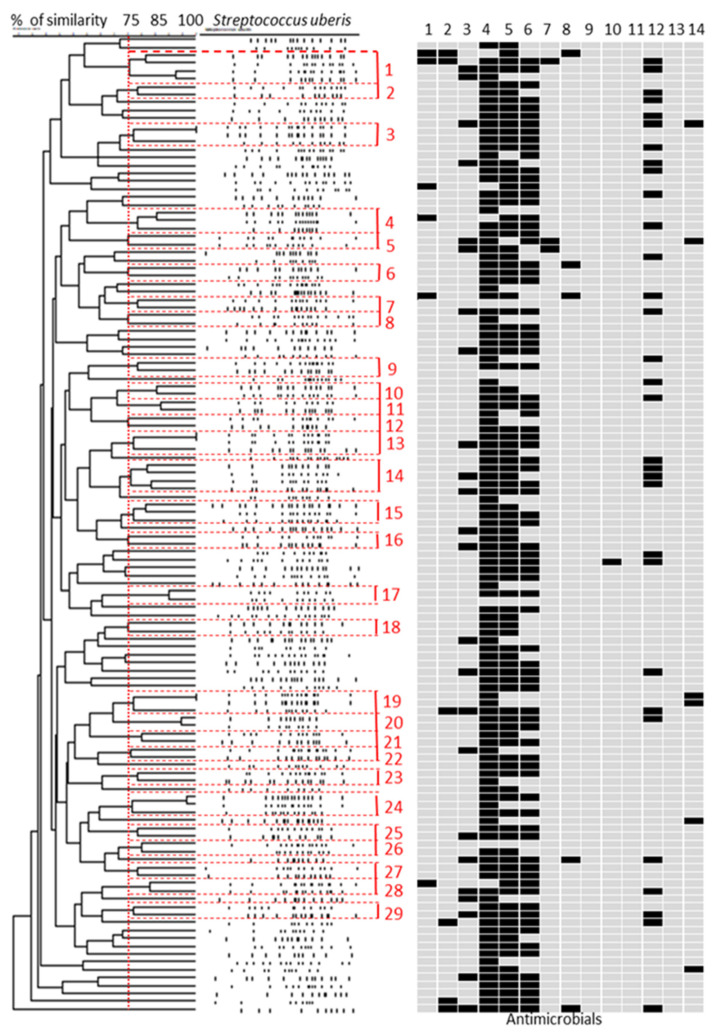
PFGE-based dendrogram of 124 *Streptococcus uberis* isolates collected from ovine milk samples and antimicrobial resistance profiles. Clusters were based on ≥75% similarity and are labeled 1 to 29 in red. For antimicrobial resistance phenotypes, black indicates resistance and grey indicates susceptible or intermediate. Antimicrobials are ampicillin (1), penicillin (2), tetracycline (3), streptomycin (4), kanamycin (5), gentamicin (6), erythromycin (7), trimethoprim-sulphamethoxazole (8), amoxicillin-clavulanic acid (9), cephalothin (10), oxacillin (11), novobiocin (12), ceftiofur (13) and pirlimycin (14).

**Figure 4 pathogens-10-01489-f004:**
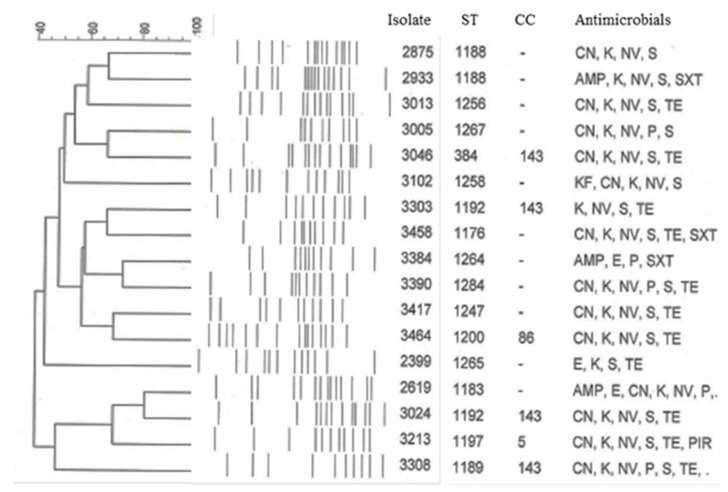
UPGMA dendrogram showing the relationship between ST, CC and antimicrobials resistance profile of the 17 MDR *Streptococcus uberis* isolates.

## Data Availability

All data generated or analyzed during this study are includes in this published article and its Appendix A.

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
