# Peer review of "Molecular Typing and Antimicrobial Susceptibility Profiles of Streptococcus uberis Isolated from Sheep Milk"

_pathogens, 2021, doi:10.3390/pathogens10111489_

Round 1

Reviewer 1 Report

This study explored the genotypic profile and antimicrobial resistance in S. uberis isolated from ovine mastitis. My comments appeared below for the author to consider to improve their manuscript:

  1. Antimicrobial resistance of S. uberis was high for streptomycin, kanamycin, gentamicin. What ST/CC/PFGE is correlated to aminoglycoside resistance? The author can describe in the result & discussion.
  2. Why S. uberis strains in this study were resisted to aminoglycoside in high frequency? Extensive used of this antibiotic in that area? or this bacteria is intrinsic resistance to this antibiotic? or  due to another aminoglycoside resistance genes? The author should discuss this mention.
  3. Novobiocin was also high detected. Can author discuss this point? Is it related to some ST/CC/PFGE?
  4. CLSI used in this study was 2018, it is not update version. Sometime the clinical breakpoints may change in the update version. The author need to check.
  5. What is STs related to MDR?
  6. Inclusion of ST/CC to PFGE or vice versa will be better to gain the best picture of S. uberis genetic diversity.

Author Response

We thank the reviewers for their suggestions. Based on their indications, we have reviewed our manuscript for clarity, and worked on the contents to increase result understandability.

Reviewer #1

  1. Antimicrobial resistance of S. uberis was high for streptomycin, kanamycin, gentamicin. What ST/CC/PFGE is correlated to aminoglycoside resistance? The author can describe in the result & discussion. Authors (AU): We have inserted a sentence in the Results section
  2. Why S. uberis strains in this study were resisted to aminoglycoside in high frequency? Extensive used of this antibiotic in that area? or this bacteria is intrinsic resistance to this antibiotic? or  due to another aminoglycoside resistance genes? The author should discuss this mention. AU: We have added a sentence in the Discussion section
  3. Novobiocin was also high detected. Can author discuss this point? Is it related to some ST/CC/PFGE?  AU: We did not find a correlation between ST /CC/PFGE and resistance to this antimicrobial.
  4. CLSI used in this study was 2018, it is not update version. Sometime the clinical breakpoints may change in the update version. The author need to check. AU: When we analyzed the Streptococcus uberis isolates, the CLSI manual published in January 2018 was available. Two notices were subsequently published, on 18 June 2018 and 27 March 2019, respectively. The first notice concerned the carbapenemase activity in Enterobacteriaceae while the second the quality control protocol.
  5. What is STs related to MDR?  AU: We have inserted the Figure 4 that links PFGE/ST/CC to antimicrobial resistance. However, we did not found any correlation  
  6. Inclusion of ST/CC to PFGE or vice versa will be better to gain the best picture of S. uberis genetic diversity. AU: To gain a better picture of genetic diversity of MDR S. uberis, we have added the Fig. 4

Reviewer 2 Report

Aim of the present study was the analysis of genetic diversity and the susceptibility to antimicrobials in a collection of Streptococcus uberis isolated from shep milk in Sardinia between January 2011 and December 2017. The methods were well done and the results are clearly described. However I have some points which should be answered. Main comments: 1. Correlation between the antimicrobial resistance and STs is important and could be added, Are some STs y more resistant than others? 2. Correlation between STs and PFGE clusters is also important. Prvious studies have shown that strains with the same PFGE profil belonged to different STs, whereas, strains of the same ST had different PFGE profils,indicating the genetic events which occur among the isolates. 3. I believe that it is not correct to test isolates for susceptibility to novobiocin, which is an agent not used for therapy. Novobiocin is only used for differentiate Staphylococcus saprophyticis from other staphylococci. 4. All streptococci have a low intrinsic resistance to aminoglycosides (www.eucast.com). Please add some comment into the text. Minor comments Page 2, Lines 83-85: please add the STs which are grouped in each Clonal complex. I believe that Table S1 must added in the main article. Page 6, Line138: what is the meaning '' seven isolates were non-typeable''. Please rephrase it Page 10: the table A1 could be included as Suppl. material

Author Response

We thank the reviewers for their suggestions. Based on their indications, we have reviewed our manuscript for clarity, and worked on the contents to increase result understandability.

Reviewer #2

  1. Correlation between the antimicrobial resistance and STs is important and could be added, Are some STs y more resistant than others? Authors (AU): We have inserted a sentence in the Results section
  2. Correlation between STs and PFGE clusters is also important. Prvious studies have shown that strains with the same PFGE profil belonged to different STs, whereas, strains of the same ST had different PFGE profils,indicating the genetic events which occur among the isolates.  AU: Please, could you tell me which part of the "discussion" section, lines 173-192 is not clear?
  3. I believe that it is not correct to test isolates for susceptibility to novobiocin, which is an agent not used for therapy. Novobiocin is only used for differentiate Staphylococcus saprophyticis from other staphylococci. AU: In this study, we analyzed Streptococcus uberis isolates with the same antimicrobial panel used for the staphylococcal strains isolated from ovine mastitis. The aim was to compare the AMR in different bacteria responsible for mastitis. For us, novobiocin resistance in 25% of the isolates was a real surprise. We believe this represents an important result

All streptococci have a low intrinsic resistance to aminoglycosides (www.eucast.com). Please add some comment into the text. AU: We have inserted a sentence in the Discussion section

Minor comments Page 2, Lines 83-85: please add the STs which are grouped in each Clonal complex. AU: We have added the Supplementary Table S2 “Distribution of Sequence Types (STs) within the three defined Clonal Complex (CC)”

I believe that Table S1 must added in the main article. AU: The Table S1 is too long to fit into the main text

Page 6, Line138: what is the meaning '' seven isolates were non-typeable''. AU: We have changed the sentence.

Please rephrase it Page 10: the table A1 could be included as Suppl. Material. Au: We have inserted the Table A1 as Supplementary Table S3

Reviewer 3 Report

The present work consists of analyzing, from the point of wiew of genetic diversity and resistance to antibiotics, 124 Streptococcus uberis isolates from sheep milk samples collected in Sardinia, Italy.

The work is interesting since it addresses an important problem from the point of view of animal welfare and the repercusssions it has for the farmer. Mastitis is a serious infectious disease tha affects the mammary glands and is very common amongst dairy-producing animals.

Treatmenta of the disease through the application of antibioticas in recents years has made environmental pathogens, such as S. uberis, wich has now become one of the main causes of mastitis in cattle and sheep mainly.

The present paper is novel since it focuses on analyzing sheep's milk isolates, which is where the role of this microorganism is less studies when compared to the studies that exist in cattle.

The authors found great variability in the lineage of the strains studied, but nevertheless the behavieor in terms of resistance to antibiotics is similar. This study opens the doors to future research to relate the types of circulating S. uberis wht the severity of the disease to find out which type is more patogenic. This great diversity of circulating types could make it difificult to develop a possible vaccine for this microorganism.

One aspect of improvement in this work would be to explain more clearly the origin of the analyzed samples. It is not clear in the paper if these milk samples were taken from diseased or healthy eves, if the severtiy of the illness was the same, or they were asymtomatic. But ultimately it is a study that gives rise to more interesting future research, but I understand that the objectives of said researcha are not for this study, but for future ones.

Author Response

We thank the reviewers for their suggestions. Based on their indications, we have reviewed our manuscript for clarity, and worked on the contents to increase result understandability.

Reviewer #3

One aspect of improvement in this work would be to explain more clearly the origin of the analyzed samples. It is not clear in the paper if these milk samples were taken from diseased or healthy eves, if the severtiy of the illness was the same, or they were asymtomatic. But ultimately it is a study that gives rise to more interesting future research, but I understand that the objectives of said researcha are not for this study, but for future ones.

AU: The study does not have an epidemiological aim but was carried out on the isolate collection obtained along the years from our routine laboratory. However we agree with the reviewer on the importance of correlating the clinical status of the animal to the genetic characterization of Streptococcus uberis isolates.

Reviewer 4 Report

The epidimological data for the 124 isolates in material and methods (line 221) are limited. 

figure 3 is not easy to understand and to extract further informations

Author Response

We thank the reviewers for their suggestions. Based on their indications, we have reviewed our manuscript for clarity, and worked on the contents to increase result understandability.

Reviewer #4

The epidimological data for the 124 isolates in material and methods (line 221) are limited.

figure 3 is not easy to understand and to extract further informations

AU: The study does not have an epidemiological aim but was carried out on the isolate collection obtained along the years from our routine laboratory.

Round 2

Reviewer 2 Report

In the revised version all comments have been  answered.